# Oleic Acid Protects Endothelial Cells from Silica-Coated Superparamagnetic Iron Oxide Nanoparticles (SPIONs)-Induced Oxidative Stress and Cell Death

**DOI:** 10.3390/ijms23136972

**Published:** 2022-06-23

**Authors:** Neža Repar, Eva Jarc Jovičić, Ana Kump, Giovanni Birarda, Lisa Vaccari, Andreja Erman, Slavko Kralj, Sebastjan Nemec, Toni Petan, Damjana Drobne

**Affiliations:** 1Biotechnical Faculty, University of Ljubljana, 1000 Ljubljana, Slovenia; 2Department of Molecular and Biomedical Sciences, Jožef Stefan Institute, 1000 Ljubljana, Slovenia; eva.jarc.jovicic@ijs.si (E.J.J.); ana.kump@ijs.si (A.K.); toni.petan@ijs.si (T.P.); 3Jožef Stefan International Postgraduate School, 1000 Ljubljana, Slovenia; 4Elettra-Sincrotrone Trieste, 34149 Trieste, Italy; giovanni.birarda@elettra.eu (G.B.); lisa.vaccari@elettra.eu (L.V.); 5Institute of Cell Biology, Faculty of Medicine, University of Ljubljana, 1000 Ljubljana, Slovenia; andreja.erman@mf.uni-lj.si; 6Department for Materials Synthesis, Jožef Stefan Institute, 1000 Ljubljana, Slovenia; slavko.kralj@ijs.si (S.K.); sebastjan.nemec@ijs.si (S.N.); 7Faculty of Pharmacy, University of Ljubljana, 1000 Ljubljana, Slovenia

**Keywords:** superparamagnetic iron oxide nanoparticles, oxidative stress, endothelial cells, lipid droplets, oleic acid

## Abstract

Superparamagnetic iron oxide nanoparticles (SPIONs) have great potential for use in medicine, but they may cause side effects due to oxidative stress. In our study, we investigated the effects of silica-coated SPIONs on endothelial cells and whether oleic acid (OA) can protect the cells from their harmful effects. We used viability assays, flow cytometry, infrared spectroscopy, fluorescence microscopy, and transmission electron microscopy. Our results show that silica-coated SPIONs are internalized by endothelial cells, where they increase the amount of reactive oxygen species (ROS) and cause cell death. Exposure to silica-coated SPIONs induced accumulation of lipid droplets (LD) that was not dependent on diacylglycerol acyltransferase (DGAT)-mediated LD biogenesis, suggesting that silica-coated SPIONs suppress LD degradation. Addition of exogenous OA promoted LD biogenesis and reduced SPION-dependent increases in oxidative stress and cell death. However, exogenous OA protected cells from SPION-induced cell damage even in the presence of DGAT inhibitors, implying that LDs are not required for the protective effect of exogenous OA. The molecular phenotype of the cells determined by Fourier transform infrared spectroscopy confirmed the destructive effect of silica-coated SPIONs and the ameliorative role of OA in the case of oxidative stress. Thus, exogenous OA protects endothelial cells from SPION-induced oxidative stress and cell death independent of its incorporation into triglycerides.

## 1. Introduction

Superparamagnetic iron oxide nanoparticles (SPIONs) are a group of nanoparticles (NPs) that have an iron oxide core with a size ranging from 5 nm to 15 nm and can be surrounded by various coatings [1]. Due to their superparamagnetic properties, SPIONs have great potential for use in medicine as magnetic drug carriers, in the treatment with magnetic hyperthermia, as contrast agents in magnetic resonance imaging, and for imaging with superparamagnetic NPs [2,3,4]. SPIONs are generally considered safe and biocompatible, but recent studies suggest they might cause aberrant cellular responses, including DNA damage, oxidative stress, mitochondrial membrane dysfunction, changes in gene expression, and lipid peroxidation [5,6,7,8]. Therefore, a detailed understanding of SPION-cell interactions and identification of approaches to reduce the side effects caused by SPIONs could improve their applicability for clinical use.

In our study, we use silica-coated SPIONs that form a very stable colloidal suspension in water and cell culture medium. The silica surface enables the formation of strong covalent Si-O-Si bonds with alkoxysilane molecules, which can then be further covalently bound with polyethylene glycol for colloidal stability or/and targeting agents [9]. Similar silica layers on the core of iron oxide NPs can also be made with alkoxysilane coatings [10]. This is also the case with a commercial product (NanoTherm AS1; MagForce Nanotechnologies) used in clinics for thermotherapy with various human cancers [11].

SPIONs are usually administered intravenously, so the endothelial cells that line blood vessels are among the first to come into contact with them. SPIONs with a hydrodynamic diameter of between 10 and 50 nm have been reported to remain in the bloodstream for a long time (the half-life of ferumoxitol is 14 h, that of ferumoxtran is 24–30 h) [12], so the effects of SPIONs on endothelial cells and the ability of cells to cope with SPION exposure are of great importance.

Lipid droplets (LDs) are highly dynamic cytosolic organelles found in most eukaryotic cells. They vary widely in size (0.1 to 100 µm) and consist of a core of neutral lipids surrounded by a phospholipid monolayer embedded with various proteins [13]. LDs originate from the endoplasmic reticulum (ER) and can associate with most other cellular organelles through membrane contact sites [14]. The main function of LDs in the cell is to regulate lipid metabolism; in addition, they have many other functions [15]. The biogenesis of LDs is augmented in cells exposed to various conditions that induce energy or redox imbalances, suggesting that LDs are essential for the management of cellular stress [16]. One of the major ways in which LDs control cellular stress is by regulating the sequestration, storage and release of fatty acids (FA) [17]. For example, LDs control the trafficking of unsaturated fatty acids to maintain proper membrane saturation and prevent ER stress during hypoxia [18].

Many studies showed that different types of NPs, among them SPIONs, increase the number of LDs in different cell types [10,19,20,21,22,23,24,25,26,27,28,29,30,31]. There are also studies showing a decrease in LDs [32] or no effect of NPs on LDs [33]. Studies by Vesterdal et al. showed that an increase in intracellular lipid content after exposure of HepG2 cells to NPs was only observed after simultaneous exposure to exogenous oleic acid (OA) and palmitic acid [34].

Few of the studies that have examined the effect of NPs on LD turnover have also examined the mechanisms involved. Some studies point to the role of oxidative stress in lipid accumulation [19,34,35,36] whereas others suggest that it may be related to destabilization of lysosomes and dysfunction of the lipophagy pathway [37], ER stress [28,35], and activation of PPARγ metabolism.

If a high number of LDs indicates that cells are better able to withstand stress, one might assume that cells in which LD synthesis is pre-stimulated by the addition of excess lipids are better protected against stress. This was investigated by Khatchadourian and Maysinger. They demonstrated that cell survival is enhanced upon exposure to cadmium telluride (CdTe) NPs when pheochromocytoma cells (PC12) were previously exposed to OA, which stimulates LD synthesis [19]. 

OA is an omega-9 monounsaturated FA found in various animal and plant sources and is the major FA in olive oil. OA is considered to be highly resistant to oxidation and is capable of enhancing the activity of antioxidants such as tocopherols [38]. In mouse 3T3-L1 fibroblasts, OA has a protective effect against cytotoxicity induced by an oxidative stress trigger, tert-butyl hydroperoxide [39]. In rat hepatoma cells, OA prevents apoptotic cell death induced by trans 10, cis12 (t10, c12)-conjugated linoleic acid [40]. In human hepatoma cells HepG2, OA ameliorates palmitic acid-induced hepatocellular lipo-toxicity by inhibiting ER stress and pyroptosis [41]. A study by Magtanong et al. [42] showed that exogenous OA inhibits ferroptotic cell death by reducing lipid peroxidation at the plasma membrane by displacing readily oxidizable polyunsaturated FAs in phospholipids. The effect of exogenous OA was not dependent on the turnover of LD but was instead based on its direct acyl-CoA synthetases 3 (ACSL3)-mediated incorporation into phospholipids. In contrast, in breast cancer cells exposed to lethal levels of exogenous polyunsaturated FA, incorporation of OA into triglycerides (TAG) and its lipolytic release from LDs was required for the prevention of oxidative stress and cell death [43]. 

The aim of our study was to investigate whether silica-coated SPIONs affect endothelial cell viability, oxidative stress, and LD accumulation. We examined the ability of exogenous OA to alter SPION-induced cell damage. Our results show that silica-coated SPIONs are internalized by human umbilical vein endothelial cells (HUVEC), where they induce oxidative stress, cause cell death, and induce accumulation of LDs. Exogenous OA protects endothelial cells from SPION-induced oxidative stress and cell death independent of its incorporation intoTAG.

## 2. Results and Discussion

### 2.1. Particle Characteristics and Their Internalization by Endothelial Cells

The physicochemical properties of NPs(size, shape, surface charge, and consequently aggregation status) affect cellular uptake, intracellular transport, and cellular response. To better understand and predict their behavior, we characterized our NPs in detail before starting experiments on cells.

The average size of the silica-coated SPIONs, as determined by transmission electron microscopy (TEM) (particles counted > 100), was 19.3 ± 2.0 nm (Figure 1A,B). The thickness of the silica shell was approximately 4 nm. The measured zeta potential value in the serum-free cell culture medium was −18.7 ± 1.6 mV and dynamic light scattering (DLS) measurements revealed an average hydrodynamic diameter of 57 nm (Figure 1C,D). The X-ray diffraction (XRD) of the precipitated NPs showed a single spinel phase, whereas the Mossbauer spectroscopy confirmed that the core of silica-coated SPIONs were composed of maghemite, as reported in our previous work [44]. More details about NPs are available in our papers published elsewhere [10,45].

Analysis of NP internalization at the cellular level is necessary for a realistic assessment of their effects. Here we used TEM, which has proven to be a powerful tool to obtain information on the uptake of NPs.

TEM examination of HUVEC cells exposed to silica-coated SPIONs revealed that silica-coated SPIONs were endocytosed after 24 h of exposure and at a concentration of 50 µg/mL. In cells, silica-coated SPIONs were largely observed in endocytic vesicles and autophagosomes (Figure 1E,F).

### 2.2. Silica-Coated SPIONs Decrease Cell Viability and Induce Elevation of ROS

Many studies have already investigated the effect of SPIONs on endothelial cells and have come to very different conclusions. Some studies have shown cytotoxic effects of SPIONs [46,47]. Other studies have shown that although SPIONs do not affect cell viability, they cause some other effects, such as decreased endothelial integrity, decreased NO synthesis, and structural changes in cells [48], decreased cell adherence [49], altered cell morphology [50], and an increase in cellular ferritin concentration [51]. On the other hand, some studies have shown that SPIONs have no statistically significant effect on any of the measured parameters in endothelial cells [52,53]. Uncoated SPIONs are colloidally unstable in aqueous media. Therefore, different types of coatings are used, which have a great impact on their physicochemical properties and consequently on their interaction with the cell. In the studies described above, SPIONs with different coatings were tested, but none of the studies investigated the effect of silica-coated SPIONs. Therefore, at the beginning of our work, we first tested the effect of our silica-coated SPIONs on cell viability.

Three spectrophotometric methods and one flow cytometric method were used to determine cell viability after exposure to silica-coated SPIONs. Each of the three spectrophotometric methods determines cell viability using a different principle. The resazurin assay, which is based on the mitochondrial activity of cells, and the neutral red uptake (NRU) assay, which assesses the ability of cells to maintain acidic pH in lysosomes, showed cytotoxicity at concentrations above 25 µg/mL after 24 h of exposure (Figure 2A,B). The CyQuant Direct Cell Proliferation Assay kit, which measures DNA content, showed cytotoxicity at concentrations above 10 µg/mL (Figure 2C). Longer exposure times did not significantly increase cytotoxic effects, but the resazurin assay and NRU assay showed cytotoxicity at concentrations above 10 µg/mL after 48 and 72 h of exposure, respectively (Figure 2A–C).

Since the three viability assays at different exposure times did not show that a longer exposure time (48 and 72 h) would significantly affect cytotoxicity, cells were exposed to silica-coated SPIONs for 24 h in all further experiments. The exposure time of 24 h might be relevant in vivo, since the half-life of SPIONs in clinical use (with a hydrodynamic diameter between 10 and 50 nm) is typically between 1 and 36 h [12,54]. In the following experiments, cell death was measured by flow cytometry using the 7-AAD dye, which binds nucleic acids and is readily taken up by dying cells, but does not enter live cells. Because flow cytometry allows multiparametric analysis, the dye 7-AAD was used for dead cell discrimination in combination with other dyes (BODIPY 493/503 and CM-H_2_DCFDA) in the following experiments. After a 24-h exposure, the percentage of dead cells at a SPION concentration of 50 μg/mL, as determined by 7-AAD, was approximately 60% (Figure 2D), which is comparable with decreased cell viability determined by resazurin assay and the NRU assay. 

Many articles have pointed out that the main mechanism of toxicity of iron oxide NPs is the formation of ROS and the resulting oxidative stress [55,56,57,58]. To determine whether silica-coated SPIONs induce ROS production and whether oxidative stress is the main cause of SPION-induced cell death, cells were simultaneously exposed to silica-coated SPIONs and one of the antioxidants (N-acetylcysteine (NAC) (20 mM) or alpha-tocopherol (1 mM)). Intracellular ROS production was measured by flow cytometry using CM-H_2_DCFDA staining, and cell death was determined using the 7-AAD dye. Our results showed that silica-coated SPIONs increase the amount of ROS in a concentration-dependent manner (Figure 2E), which is consistent with literature reports suggesting that the main mechanism of toxicity of iron oxide NPs, including SPIONs, is the generation of ROS and the resulting oxidative stress [56,57,58,59,60,61]. Moreover, we found that both NAC and α-tocopherol decreased the amount of ROS (Figure 2E) and reduced the cell death caused by silica-coated SPIONs (Figure 2D), suggesting that the cytotoxic effect of silica-coated SPIONs is indeed dependent on oxidative stress. The reduction in the amount of ROS and the proportion of cell death was statistically significant only for NAC with the higher silica-coated SPION concentration (50 µg/mL).

### 2.3. Silica-Coated SPIONs Induce LD Accumulation 

Oxidative stress affects cellular lipid metabolism and could lead to the accumulation of LDs in cultured cells [62,63]. Some studies have suggested that NPs stimulate the formation of LD via oxidative stress [19,34,35,36]. Given the SPION-induced increase in ROS in HUVEC cells, we asked whether silica-coated SPIONs also induce LD biogenesis. To test the hypothesis, cells were treated with silica-coated SPIONs (25, 50, and 100 µg/mL) in serum-free medium and LD content in live cells was determined after 24 h. We observed a higher number of LDs in SPION-treated cells compared to untreated cells (Figure 3A), suggesting that silica-coated SPIONs induce LD accumulation. To further determine whether the cause for LD accumulation is increased LD synthesis or decreased LD degradation, cells were simultaneously exposed to silica-coated SPIONs and LD synthesis inhibitors (DGAT inhibitors: T863 (5 µM) and PF-06424439 (5 µM)). LDs and cell death were quantified by flow cytometry using BODIPY 493/503 and 7-AAD staining, respectively. The results showed that the addition of LD synthesis inhibitors does not affect the number of LDs induced by silica-coated SPIONs (Figure 3B), or the proportion of cell death caused by silica-coated SPIONs (Figure 3C), implying that silica-coated SPIONs do not induce LD synthesis but inhibit their degradation. To further test this hypothesis, we determined the number of LDs over a longer period of time (96 h) in silica-coated SPION-treated cells. The results show that the number of LDs in both control and silica-coated SPION-treated cells decreases over time, with a slower decrease in cells exposed to silica-coated SPIONs (Figure 3D), which confirms our previous results, showing that silica-coated SPIONs inhibit LD degradation. At first glance, it may seem that our results are not consistent with the results of numerous studies that have shown that NPs induce LD synthesis [19,20,21,22,23,24,25,26,27,28,29]. However, most of these studies only showed that cells exposed to NPs had a higher amount of LD, and then explained that NPs induce the synthesis of LD.

### 2.4. LD Formation Is Not Required for OA-Induced Inhibition of Silica-Coated SPION-Induced Oxidative Stress and Cell Death

LDs may have a protective role against oxidative stress [15,64,65,66]. To test the hypothesis that cells with more LDs are better protected from the deleterious effects of silica-coated SPIONs than those with less LDs, OA treatment was used to allow HUVEC cells to form an LD pool prior to exposure to silica-coated SPIONs. In our experiments, we used 30, 50, and 100 µM OA, which corresponds to an in vivo relevant concentration, since plasma concentrations of oleate are likely to be between 10 and 100 µM under conditions of high olive oil consumption [67]. LD formation was confirmed by flow cytometry (Figure 4A) and microscopy (Figure 4D). As expected and consistent with literature reports [67,68,69], we confirmed that OA induces the formation of LDs in HUVEC cells in a dose dependent manner (Figure 4A). 

In order to test whether OA provides a survival advantage for HUVEC cells upon exposure to silica-coated SPIONs, HUVEC cells were pretreated with OA (100 µM) to form a LD pool and then treated with silica-coated SPIONs (25 and 50 µg/mL) for 24 h. Pretreatment of HUVEC cells with OA increased their resistance to cell death induced by silica-coated SPIONs (Figure 4B), and the protective effect was associated with suppression of ROS (Figure 4C), consistent with previous studies showing a protective role of OA against oxidative stress and cell death [39,40,41,42]. Another study has also shown the protective effect of OA in cells treated with NPs; OA treatment against metallic CdTe NPs-induced oxidative stress improves cell survival and prevents lysosome enlargement in pheochromocytoma cells (PC12) [19].

Given that excess FAs may be toxic to cells, we hypothesized that when OA is added exogenously, DGAT-mediated formation of LDs is induced to prevent OA lipo-toxicity. To test this idea, we inhibited DGAT-mediated TAG synthesis with the DGAT1 and DGAT2 inhibitors T863 and PF-06424439, respectively, and simultaneously exposed the cells to OA. As expected, the DGAT inhibitors almost completely inhibited OA-induced formation of LDs (Figure 4D,E) and caused OA lipo-toxicity in HUVEC cells at higher OA exposure concentrations (50 and 100 µM) (Figure 4F). This means that cells can tolerate OA in its free form at lower concentrations (30 µM and less). At higher concentrations (50 μM and more), the free OA would become toxic to the cells, so the cells store it in the form of LDs.

To test whether the formation of LDs is a prerequisite for exogenous OA to inhibit cell death caused by silica-coated SPIONs, we performed a series of experiments with a combination of LD formation inhibitors (DGAT inhibitors), silica-coated SPIONs, and OA. We used a lower concentration of OA (30 µM) because we had previously shown that this concentration could be used together with DGAT inhibitors without causing lipo-toxicity (Figure 4F). Surprisingly, the results showed that 30 µM OA was still able to suppress cell death induced by silica-coated SPIONs in the presence of DGAT inhibitors (Figure 4G), suggesting that the pro-survival effect of OA does not depend only on its incorporation into LDs.

Our results are consistent with the study by Magtanong et al., in which they showed that the protective effect of monounsaturated FAs against a type of cell death (ferroptosis) does not depend on the formation of LDs but on the displacement of easily oxidizable polyunsaturated fatty acids (PUFAs) from membrane phospholipids and the simultaneous suppression of membrane lipid peroxidation [42].

### 2.5. Fourier Transform Infrared (FTIR) Spectroscopy Confirmed That OA Attenuates the Negative Effects of Silica-Coated SPIONs and Revealed Additional Mechanisms of Silica-Coated SPION Cytotoxicity

To test the mechanism of cytotoxicity induced by silica-coated SPIONs and the attenuation of the negative effects by OA, untreated and treated cells were analyzed by FTIR spectroscopy. The data set for the infrared (IR) measurements included more than 700 spectra (738 analyses of individual cells), which were divided into four classes: CTRL (untreated cells), SPIONs (cells treated for 24 h with silica-coated SPIONs at a concentration of 50 µg/mL), OA (cells treated for 24 h with OA at a concentration of 100 µM), and OA + SPIONs (cells treated for 24 h simultaneously with OA (100 µM) and silica-coated SPIONs (50 µg/mL)).

#### 2.5.1. Univariate Analysis: Silica-Coated SPIONs Decrease Cellular Metabolism and Membrane Plasticity 

The first approach was to analyze how the treatment affects the specific IR bands and the intensity of the spectra. The intensity of the average spectra, indicating the overall cellular content, is higher for CTR, OA, and OA + SPIONs and lower for SPIONs (Figure 5A). In particular, the band at 1100 cm^−1^, which can be attributed to the symmetric phosphate stretching of nucleic acids, is less pronounced in the SPION group compared to the other treated cells, indicating a possible lower content of nucleic acids in SPION treated cells.

A closer look at the spectral bands reveals that SPIONs have a red shift of the lipid’s alkyl chains’ CH_2_ main peaks (at 2925 cm^−1^ and 2855 cm^−1^) of about 3–4 cm^−1^. The bands at 2925 cm^−1^ and 2855 cm^−1^ correspond to CH_2_ symmetric stretching and CH_2_ asymmetric stretching, respectively [70]. Observed change is just at the limit of the spectral resolution, but still indicative of increased membrane stiffness in cells treated with silica-coated SPIONs. 

In addition, only in SPIONs is band broadening observed in the amide I region, i.e., an increase in bands at 1632 cm^−1^ is seen in the second derivative analysis. This change can be attributed to β-sheet protein conformation, which is often caused in cells by unhealthy conditions such as oxidative stress or pre-apoptosis [71].

A comprehensive picture of the cellular state is obtained by observing the trends in the ratios of the selected bands (Figure 5B–D). The nucleic acid/protein ratio (Figure 5B), which can be considered as an indicator of the cellular metabolism and viability, is the highest in CTRL, with OA and OA + SPIONs showing similar values; ratios are significantly lower in SPIONs. In contrast, the lipid/protein ratio (Figure 5C), index of the unbalance in the cellular metabolism or higher lipid content, has an almost opposite trend, with CTRL cells having the lowest value, whereas SPIONs or OA + SPIONs have the highest value. OA are in an intermediate state and have a mean value that is statistically different from that of CTRLs and SPIONs. The CH_3_/CH_2_ ratio (Figure 5D) focuses on the composition of the aliphatic chains of the lipids, in particular the branching/length of unsaturated lipid chains. High CH_3_/CH_2_ levels can be associated with higher levels of cell membrane plasticity and fluidity [72]. Low CH_3_/CH_2_ levels, on the other hand, may be associated with lipid peroxidation [72], oxidative stress [73], cell death [74], and membrane blebbing and vesicle formation in apoptotic cells [71]. Moreover, the increase in CH_2_ and consequently reduced CH_3_/CH_2_ ratio can be assigned to an increase in the amount of long-chain PUFAs that could be linked to the apoptosis, necroptosis [75,76], or ferroptosis [77]. The CH_3_/CH_2_ ratio is the highest for CTRLs, followed by OA, OA + SPIONs and it has the lowest value for SPIONs.

#### 2.5.2. PCA Analysis: Silica-Coated SPIONs Cause Variation in Secondary Protein Structure and DNA Damage

To observe in more detail how the overall population of cells responded to the different treatments, principal component analysis (PCA) analysis was performed on the collected data sets. The scatter plot (Figure 6A) show how the spectra of the cells are distributed in the PCA plane identified by principal component (PC) 1 and 4, which are the components that provide the best separation between the four groups. The CTRLs, in black, have a narrow distribution with the barycenter almost coinciding with the (0,0) coordinate. Then it is possible to see that PC4 separates the OA and OA + SPIONs from SPIONs, while along PC1 there is a separation of OA + SPIONs in the positive hemi plane and OA in the negative hemi plane. For both, OA + SPIONs and OA, almost half of the populations differ from the controls along PC1.

By combining the values of the scores shown in Figure 6A with the spectral features identified by the loading vectors in Figure 6B, it is possible to assign the spectral variation to the biochemical features that caused the separation.

Beginning in the higher wavenumber region, PC1 is characterized by three sharp negative peaks at 2923 cm^−1^ and 2851 cm^−1^ assigned to CH_2_ stretching in an ordered/packed state and one at 2858 cm^−1^ assigned to CH_3_. The intensity of both CH_3_ and CH_2_ signals increases along PC1 from left to right, i.e., from OA to CTRL to OA + SPIONs, indicating an increase in lipid order/stiffness, which can be related to the formation of LDs [78].

PC4 shows two strong positive signals at 2929 cm^−1^ and 2859 cm^−1^ in this spectral region, which are blue-shifted by 8 cm^−1^ compared to the PC1 signals and are assigned to a lipid component with loose lipid content. OA, OA + SPIONs, and CTRLs show similar values along the PC4 plane, while SPIONs tend more toward the positive hemi plane. Since the analyzed data are second derivatives, a progressive increase in membrane fluidity from the positive hemi-plane to the negative one can be seen, indicating a lower membrane fluidity in SPIONs.

PC1 in the amide I region of the proteins is characterized by four strong signals: a positive one at 1663 cm^−1^ assigned to turns, and three negative ones: at 1705 cm^−1^ that could be assigned to β-turns, 1646 cm^−1^ assigned to random turns/loops, and 1620 cm^−1^ assigned to the parallel β-sheet [79]. Thus, along PC1 (from left to right) we see a progressive increase in parallel β-sheets, with a small decrease in random coil structures.

The strongest PC4 signals in the same region are 1713 cm^−1^, which could be assigned to the C=O of lipids, 1696 cm^−1^ and 1632 cm^−1^, both assigned to β-sheets, and 1680 cm^−1^ assigned to coils. The present variation in these signals could be due to the general decrease in protein content. Combining the signals from PC and the average spectra, we can say that PCA detects no change in the α-helices conformation at 1654 cm^−1^, while we can see a transition between turns and β-sheet conformations. From these data, it appears that the proteins in SPIONs vary the most, with a decrease in total protein content (from the spectra) accompanied by a transition to β-aggregates and random structures. The total protein content of OA + SPIONs and CTRLs remains similar. Cells treated with OA show a slight decrease in total protein content associated with an increase in random coils.

In the lower range of components (1300–900 cm^−1^), PC1 does not show intense signals, while PC4 shows three sharp negative peaks. The one at 1140 cm^−1^, observed also in the analysis of the second derivative of the mean values, shows only in SPIONs. A main peak at 1088 cm^−1^, attributed to the symmetric stretching of the R-OPO_3_^2-^ chemical groups of DNA, is present in all treatments, although we have to recall that SPION cells have the lowest number of nucleic acids when analyzing the average spectra. The signal at 1052 cm^−1^ of the C-O groups of the DNA skeleton present in CTRL, OA, OA + SPIONs and splits into two signals at 1059 cm^−1^ and 1048 cm^−1^ for SPIONs, probably because of the low signal-to-noise ratio due to the lower amount of nucleic acids. The symmetric stretching of DNA R-OPO_3_^2−^ chemical groups, i.e., the changes in the spectral region of the DNA backbone (950–1240 cm^−1^), could be related to DNA damage such as single and double strand breaks, DNA–DNA, and DNA–protein crosslinks [80].

#### 2.5.3. Spectral Phenotype of Cells

As expected, we found that cells treated with OA were most similar in spectral profile to untreated cells. Treatment with silica-coated SPIONs significantly altered the molecular profile of exposed cells, pointing to cytotoxic effects. The molecular profile of cells treated simultaneously with OA and silica-coated SPIONs was somewhere between the molecular profile of untreated cells and cells treated with silica-coated SPIONs alone in terms of their properties. This confirms our previous results showing that OA protects cells from the negative effects of silica-coated SPIONs.

Univariate and PCA analysis of FTIR measurements revealed that silica-coated SPIONs increase cellular metabolic imbalance and oxidative stress, leading to decreased cell viability. These results are consistent with our viability assays and flow cytometry results showing that silica-coated SPIONs induce ROS formation, decrease cell metabolic activity, and induce cell death. The FTIR measurements also showed that OA reduced all of the above cellular changes caused by silica-coated SPIONs, which also confirmed our previous results indicating the protective role of OA.

Moreover, the FTIR also revealed additional cytotoxic mechanisms of silica-coated SPIONs. Both univariate and PCA analysis of the FTIR measurements confirmed that silica-coated SPIONs decreased cell membrane fluidity. In addition, the low CH_3_/CH_2_ ratio in cells exposed to silica-coated SPIONs suggests the possibility that silica-coated SPIONs cause lipid peroxidation of cell membranes. Similar results were shown by Wu et al. [7] who showed that ultrasmall SPIONs catalyze the Fenton reaction to produce ·OH and cause lipid peroxidation due to the release of Fe^2+^ in human breast cancer cells. OA was able to reduce the effects of silica-coated SPIONs on cell membranes, which is consistent with the study by Magtanong et al. [42] showing that OA reduces the sensitivity of plasma membrane lipids to lipid peroxidation and resulting ferroptosis. Moreover, the observed changes in the spectral region of the DNA backbone could indicate DNA damage.

## 3. Materials and Methods

### 3.1. Materials

HUVEC were a kind gift of Špela Zemljič Jokhadar (Institute of Biophysics, Medical Faculty, University of Ljubljana, Ljubljana, Slovenia). Dulbecco’s Modified Eagle Medium (DMEM) high glucose, fetal bovine serum (FBS), Hanks’ balanced salt solution (HBSS), Dulbecco’s phosphate buffered saline (DPBS), L-glutamine, fatty acid free bovine serum albumin (FAF-BSA), NAC, α-tocopherol, resazurin sodium salt, NR, 7-aminoactinomycin D (7-AAD), Hoechst 33,342, T863, PF-06424439, cacodylic acid, sodium hydroxide, ethanol, acetic acid, sucrose, osmium tetroxide, formaldehyde, and glutaraldehyde were obtained from Sigma Aldrich (St. Louis, MO, USA), TrypLE Select and CyQuant Direct Cell Proliferation Assay Kit from Life Technologies (Carlsbad, CA, USA). OA was from Cayman Chemical (Ann Arbor, MI, USA), BODIPY 493/503 and CM-H_2_DCFDA from Thermo Fisher Scientific (Waltham, MS, USA). 

### 3.2. Synthesis and Characterization of Silica-Coated SPIONs

Silica-coated SPIONs were synthesized by coprecipitation from aqueous solutions of iron salts, as described in detail elsewhere [81,82,83]. Then, the surface of the NPs was functionalized by addition of citric acid and coated with a ~4 nm thick silica shell as described in our published studies [84,85,86,87]. The silica-coated SPIONs were characterized by TEM, DLS, and zeta potential measurements. Samples for TEM analysis were prepared by drying the aqueous suspension of silica-coated SPIONs at room temperature on a transparent carbon film on a copper grid, and images were acquired using a JEOL 2100 microscope (JEOL Ltd., Tokyo, Japan). Suspension of silica-coated SPIONs in serum-free medium (c = 1 mg/mL; pH = 7.91) was monitored by electrokinetic measurements of zeta potential (Anton Paar GmbH, Litesizer 500, Graz, Austria) and hydrodynamic diameter distribution was determined by DLS (Anton Paar GmbH, Litesizer 500, Graz, Austria).

### 3.3. Cell Culture Conditions and Treatments

HUVEC cells were cultured in Dulbecco’s Modified Eagle Medium (DMEM) supplemented with 10% *v*/*v* FBS and 4 mM L-glutamine (complete medium). Cells were cultured in a humidified atmosphere with 5% CO_2_ at 37 °C and were routinely sub-cultured twice per week. For experiments, cells were harvested with TrypLE Select and plated in complete medium at a seeding density of 3 × 10^4^ cells/cm^2^. They were allowed to attach for 24 h and then OA (30 µM, 50 µM, or 100 µM), DGAT inhibitors (5 µM T863 and 5 µM PF-06424439), silica-coated SPIONs (10–100 µg/mL), and antioxidants (20 mM NAC and 1 mM α-tocopherol) were added in serum depleted medium. Before addition to cell culture, OA was complexed with 0.02% FAF-BSA while α-tocopherol was complexed with 0.5% FAF-BSA in serum depleted medium. In the experiments in which OA was not used, control cells were grown in serum-free medium only. In the experiments with OA, control cells were grown in the presence of 0.02% FAF-BSA in serum-free medium. DGAT inhibitors were added to cells 0.5 h before OA or silica-coated SPIONs and OA was added 2 h prior silica-coated SPIONs. 

### 3.4. TEM Analysis of the Cells

HUVEC cells at a density of 3 × 10^4^ cells/cm^2^ were seeded on Falcon™ Cell Culture Inserts (Thermo Fisher Scientific Inc., Waltham, MA, USA) in complete medium and allowed to adhere for 24 h. Cells were then treated with silica-coated SPIONs at a concentration of 50 µg/mL for 24 h in serum-free medium. After incubation, cells were washed three times with phosphate-buffered saline (PBS) and fixed with a mixture of 4% (*w*/*v*) formaldehyde and 2% (*w*/*v*) glutaraldehyde in 0.1 M cacodylate buffer (pH 7.4) for 3 h at 4 °C. Overnight rinsing in 0.33 M sucrose in 0.2 M cacodylate buffer was followed by post-fixation in 1% osmium tetroxide in 0.1 M cacodylate buffer for 1 h, followed by dehydration in graded ethanol and embedding in Epon 812 resin (Electron Microscopy Sciences, Hatfield, PA, USA). Ultrathin sections were cut with an ultramicrotome Leica EM UC6 and examined in a Philips CM100 transmission electron microscope (Eindhoven, The Netherlands) at 80 kV.

### 3.5. Resazurin Viability Assay

The resazurin viability assay provides a quantitative indication of cellular metabolic activity. Reduction of the indicator dye by cellular metabolic processes results in conversion of the blue, non-fluorescent oxidized form called resazurin to a red, fluorescent resorufin that can be detected by measuring fluorescence intensity.

Cells with density 3 × 10^4^ cells/cm^2^ were seeded in 96-well plates and incubated with silica-coated SPIONs (concentrations of 5, 10, 25, 50, 75, and 100 µg/mL) in serum-free medium for 24, 48, or 72 h after attachment. Resazurin at a final concentration of 0.025 mg/mL was added to each well. After 3 h at 37 °C, 100 µL of cell medium was transferred to black 96-well plates, and fluorescence intensity was measured using a spectrofluorometer (BioTek, Cytation 3, Bad Friedrichshall, Germany) (560/590 nm ex/em).

### 3.6. NRU Assay

The neutral red (NR) dye can be taken up by living cells and incorporated into lysosomes, while non-living cells are unable to take up the dye. Based on the amount of dye released from lysosomes using an organic solvent, the number of living cells can be determined.

Cells with density 3 × 10^4^ cells/cm^2^ were seeded in 96-well plates and incubated with silica-coated SPIONs (concentrations of 5, 10, 25, 50, 75, and 100 µg/mL) in serum-free medium for 24, 48, or 72 h after attachment. NR Solution at a final concentration of 0.04 mg/mL was added to each well. After 2 h, cells were rinsed with PBS and internalized dye was released using 120 µL of NR solvent consisting of 50% (*v*/*v*) ethanol, 1% (*v*/*v*) acetic acid, and 49% (*v*/*v*) deionized water. 100 µL of the solution was transferred to black 96-well plates and fluorescence intensity was measured (530/645 nm ex/em) using a spectrofluorometer (BioTek, Cytation 3, Bad Friedrichshall, Germany).

### 3.7. CyQUANT Direct Cell Proliferation Assay

The CyQuant Direct Cell Proliferation Assay Kit (Life Technologies, Carlsbad, CA, USA) contains two components: a dye that exhibits strong fluorescence enhancement when bound to nucleic acids and a fluorescence quencher. The fluorescent dye permeates all cells and concentrates in the nucleus, where it stains DNA. The fluorescence quencher cannot penetrate living cells, but only cells with a defective membrane, so that only healthy cells with an intact membrane fluoresce after addition of the two components.

Cells with density 3 × 10^4^ cells/cm^2^ were seeded in 96-well plates and incubated with silica-coated SPIONs (concentrations 5, 10, 25, 50, 75, and 100 µg/mL) in serum-free medium for 24, 48, or 72 h after attachment. Cell viability was quantified according to the manufacturer’s instructions.

### 3.8. LDs and Cell Death Assay with BODIPY 493/503 and 7-AAD

LDs and cell death were determined by flow cytometry using BODIPY 493/503 and 7-AAD, respectively, as previously described [88]. BODIPY 493/503 is a fluorescent dye that can be used to quantify neutral lipid content by flow cytometry and to observe LDs by microscopy. 7-AAD is a fluorescent dye that binds to double-stranded DNA. It is a membrane-impermeable dye that is generally excluded from viable cells and therefore can be used to distinguish dead cells.

Floating and adherent cells were collected and centrifuged at 300× *g* for 10 min. The supernatant was removed and cells were incubated with BODIPY 493/503 (0.5 µg/mL) in HBSS at room temperature for 10 min. After 10 min, 7-AAD was added to a final concentration of 5 µg/mL and the cells were incubated for an additional 10 min. Samples were acquired using a FACSCalibur flow cytometer (BD Biosciences). Fluorescence signals from BODIPY 493/503 and 7-AAD were detected using FL-1 (530/30) and FL-3 (650LP) filters, respectively. Samples were analyzed for 2 × 10^4^ events per sample.

### 3.9. ROS and Cell Death Assay Using CM-H_2_DCFDA and 7-AAD

ROS and cell death were determined by flow cytometry as described previously [43]. The cell-permeable CM-H_2_DCFDA diffuses passively into cells and is retained in the intracellular space after cleavage by intracellular esterases to CM-H_2_DCF. Upon oxidation by ROS, the non-fluorescent CM-H_2_DCF is converted to the highly fluorescent CM-DCF.

Floating cells and adherent cells were collected and stained with 1 μM CM-H_2_DCFDA in Hanks’ balanced salt solution (HBSS) for 30 min at 37 °C followed by 2 h recovery period in DMEM phenol red-free medium. Then, 7-AAD was added at a final concentration of 5 μM and cells were incubated for 10 min at room temperature. Samples were acquired using a FACS Calibur flow cytometer (BD Biosciences). At least 2 × 10^4^ cells per sample were analyzed by flow cytometry using FL-1 (530/30) and FL-3 (650LP) filters for CM-H_2_DCFDA and 7-AAD fluorescence, respectively.

### 3.10. Statistical Analysis

Statistical analysis for viability tests and flow cytometry experiments was performed using GraphPad Prism Software Version 6.01. Data are presented as means ± standard error of the mean (SEM). Statistical significance was determined using one-tailed ANOVA, followed by Tukey’s post-hoc test. *p*-values < 0.05 were considered statistically significant (* *p* < 0.05, ** *p* < 0.01, *** *p* < 0.001, **** *p* < 0.0001).

### 3.11. Epifluorescence Microscopy

Cytosolic LDs were visualized using the neutral lipid stain BODIPY 493/503 as described previously [88]. Cells with density 3 × 10^4^ cells/cm^2^ were seeded on coverslips and treated with OA (100 µM) and DGAT inhibitors (5 µM T863 and 5 µM PF-06424439). After 24 h, cells were washed with PBS and fixed in 4% formaldehyde at 37 °C. After fixation, cells were washed twice with PBS and stained with 1 μg/mL BODIPY 493/503 for 10 min, washed with PBS, incubated with Hoechst 33,342 staining solution (2 µg/mL) for 30 min, and washed again with PBS. Cover slips containing cells were placed on microscopic slides and examined with an epifluorescence microscope (Axio Imager.Z1; Carl Zeiss).

### 3.12. FTIR

#### 3.12.1. FTIR Measurements

Adherent cells were collected, washed with PBS and fixed in 4% formaldehyde for 30 min. After fixation, cells were washed with PBS and then resuspended in fresh PBS. IR data were acquired during beam time number 20175212 at the SISSI-Bio Branchline of the SISSI beamline at Elettra Sincrotrone Trieste. Single cell spectra were recorded under hydrated conditions using a homemade liquid cell, with single cells selected by closing the knife apertures at 20 × 20 microns. Measurements were performed using a Hyperion 3000 IR /Vis microscope (BRUKER Optics, Billerica, MA, USA) with a Mercury Cadmium Telluride (MCT) detector (Infrared Associates Inc., Stuart, FL, USA) coupled to a Vertex70 V in vacuum interferometer (BRUKER Optics, Billerica, MA, USA) and synchrotron emission as the source. The acquisition parameters were as follows: 512 scans at 4 cm^−1^ spectral resolution, interferometer scanner speed 120 kHz. 

#### 3.12.2. Processing of the FTIR Spectroscopy Data

After acquisition, the data were corrected for the contribution of atmospheric water and carbon dioxide using OPUS 8.5 (BRUKER Optics, Billerica, MA, USA ) and the buffer contribution was subtracted using a procedure written in-house. Then the data were processed in Quasar (quasar.codes), where they were cut in the range 3100–900 cm^−1^. The second derivative was calculated using the Savitzky-Golay algorithm with 21 points smoothing, and vector normalization was applied over the whole range. PCA was performed on the post-processed data sets using both the entire spectral range and that of the lipids (3100–2800 cm^−1^ and 1800–1700 cm^−1^) to single out the variations belonging to the lipid components. Band integrals corresponding to C=C (3115–3000 cm^−1^), CH_3_ (2995–2945 cm^−1^), CH_2_ (2945–2895 cm^−1^), lipids (3000–2800 cm^−1^), C=O (1760–1700 cm^−1^), proteins (1700–1480 cm^−1^), and nucleic acids (1280–1000 cm^−1^) were calculated by fitting a linear baseline between each interval. The band ratios were then calculated. The differences in the values of the integrals and the ratio were evaluated using a one-way ANOVA with Tukey and Bonferroni Tests, in Origin 2021 (OriginLab, Northampton, MA, USA.).

## 4. Conclusions

Our results contribute to the understanding that silica-coated SPIONs indeed cause cell death by oxidative stress and that OA is able to reduce the negative effects of silica-coated SPIONs. LDs are important organelles that protect against cellular stress, and our results confirm their role in protecting cells against lipo-toxicity when exposed to high levels of OA. Interestingly, we have shown that the incorporation of OA into LDs is not essential for protective function. The mechanism for OA cell protection is clearly more complex and involves not only incorporation into LDs but probably also incorporation of OA into the cell membrane, resulting in reduced membrane susceptibility to lipid peroxidation. To verify the obtained results, it would be very interesting in the future to quantify lipid peroxidation and genotoxicity in silica-coated SPION-treated cells and in cells treated simultaneously with OA and silica-coated SPIONs.

## Figures and Tables

**Figure 1 ijms-23-06972-f001:**
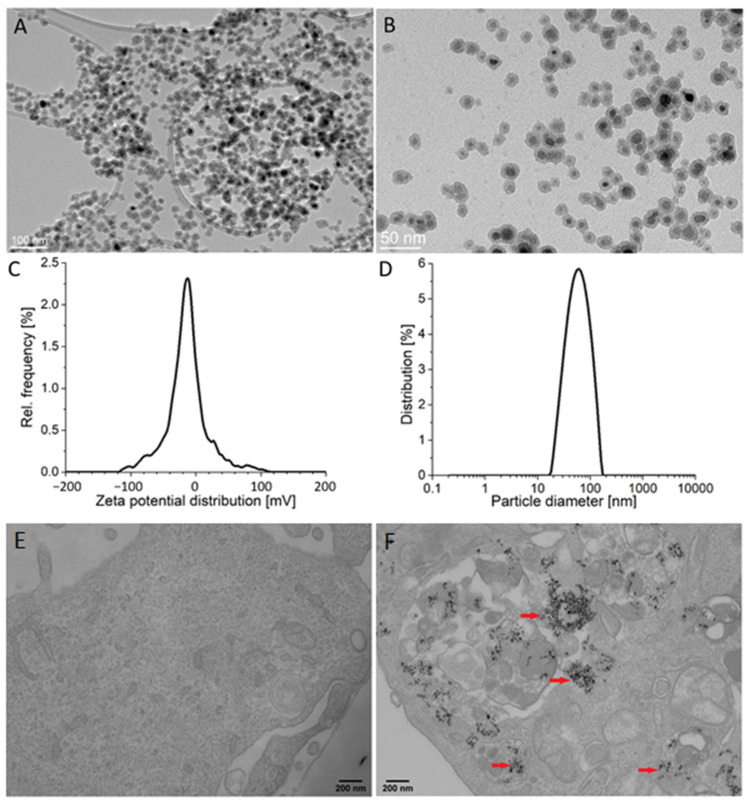
Particle characteristics and their internalization by endothelial cells. (**A**,**B**) Representative transmission electron microscopy (TEM) images of silica-coated superparamagnetic iron oxide nanoparticles (SPIONs) at low (**A**) and high magnification (**B**). (**C**) Zeta potential distribution of silica-coated SPIONs in serum-free medium. (**D**) Hydrodynamic size distribution of silica-coated SPIONs in serum-free medium. (**E**,**F**) The internalization of silica-coated SPIONs in human umbilical vein endothelial cells (HUVEC): representative TEM images of untreated HUVEC cells (**E**) and HUVEC cells exposed to silica-coated SPIONs at concentration 50 µg/mL for 24 h (**F**). Note numerous internalized electron-dense silica-coated SPIONs (red arrows) in autophagosomes.

**Figure 2 ijms-23-06972-f002:**
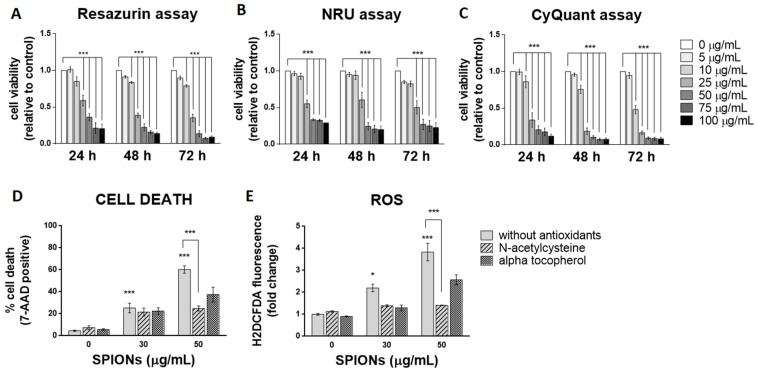
Silica-coated SPIONs decrease cell viability and induce elevation of reactive oxygen species (ROS). (**A**–**C**) HUVEC cells were treated with silica-coated SPIONs in serum-free medium for 24, 48, or 72 h. Cell viability was determined by resazurin assay (**A**), neutral red uptake (NRU) assay (**B**), and CyQuant assay (**C**). (**D**,**E**) HUVEC cells were treated with silica-coated SPIONs and N-acetylcysteine (NAC) (20 mM) or α-tocopherol (1 mM) for 24 h. Cell death (**D**) and ROS (**E**) were quantified by flow cytometry using 7-aminoactinomycin D (7-AAD) staining and CM-H_2_DCFDA staining, respectively. Values on the graphs are means ± standard error of the mean (SEM) of at least three independent experiments; statistically significant differences in mean values are indicated (*, *p* < 0.05; ***, *p* < 0.001; one-way ANOVA with Tukey post hoc test).

**Figure 3 ijms-23-06972-f003:**
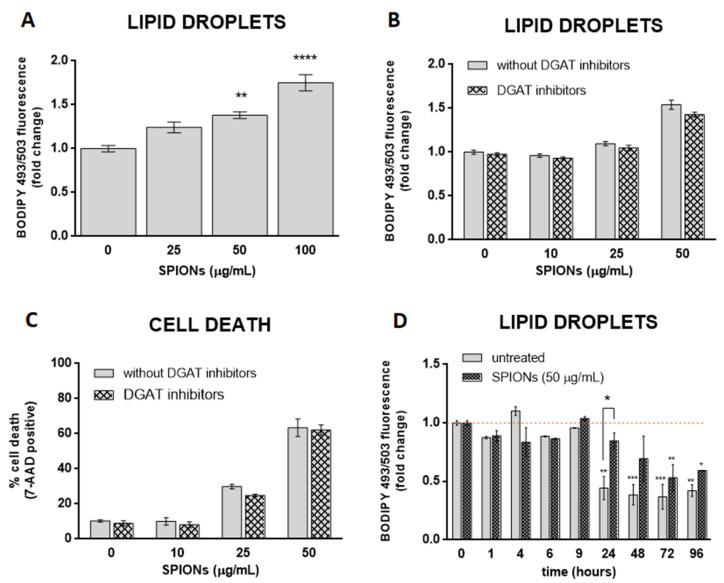
Silica-coated SPIONs-induced lipid droplet (LD) accumulation is not dependent on diacylglycerol acyltransferase (DGAT)-mediated LD biogenesis. (**A**) HUVEC cells were treated with silica-coated SPIONs (25, 50, and 100 µg/mL) for 24 h and LDs were quantified by flow cytometry using BODIPY 493/503 staining. (**B**,**C**) HUVEC cells were treated with combination of DGAT1 and DGAT2 inhibitors (5 µM T863 and 5 µM PF-06424439) and silica-coated SPIONs for 24 h. LDs and cell death were quantified by flow cytometry using BODIPY 493/503 (**B**) and 7-AAD staining (**C**), respectively. (**D**) HUVEC cells were treated with silica-coated SPIONs and LD levels were quantified in different time points (1–96 h) by flow cytometry using BODIPY 493/503 staining. Values on the graphs are means ± SEM of at least three independent experiments; statistically significant differences in mean values are indicated (*, *p* < 0.05; **, *p* < 0.01; ***, *p* < 0.001; ****, *p* < 0.0001; one-way ANOVA with Tukey post hoc test).

**Figure 4 ijms-23-06972-f004:**
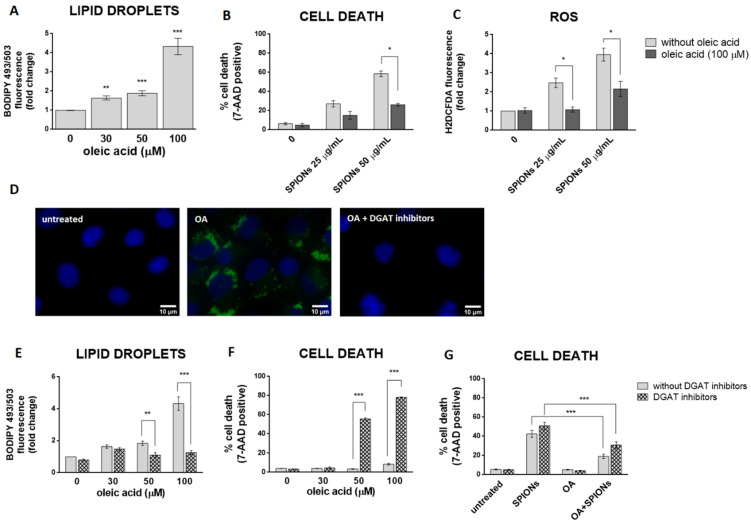
LD formation is not required for OA to inhibit silica-coated SPIONs-induced cell death and oxidative stress. (**A**) HUVEC cells were treated with 30, 50, and 100 µM OA for 24 h and LDs were quantified by flow cytometry, using BODIPY 493/503 staining. (**B**,**C**) Cells were treated with OA (100 µM) and silica-coated SPIONs (25 and 50 µg/mL) for 24 h. Cell death and ROS were quantified by flow cytometry using 7-AAD (**B**) and CM-H_2_DCFDA (**C**) staining, respectively. (**D**) Cells were treated with 100 µM OA and/or DGAT inhibitors (5 µM T863 and 5 µM PF-06424439) for 24 h. Cells were stained with BODIPY 493/503 and Hoechst 33342 to visualize LD (green) and nuclei (blue), respectively. (**E**,**F**) HUVEC cells were treated with DGAT inhibitors (5 µM T863 and 5 µM PF-06424439) and OA (30, 50, and 100 µM) for 24 h. LD and cell death were quantified by flow cytometry using BODIPY 493/503 (**E**) and 7-AAD (**F**) staining, respectively. (**G**) HUVEC cells were treated with DGAT inhibitors (5 µM T863 and 5 µM PF-06424439), OA (30 µM), and silica-coated SPIONs (50 µg/mL) for 24 h. Cell death was quantified by flow cytometry using 7-AAD staining. Values on the graphs are means ± SEM of at least three independent experiments; statistically significant differences in mean values are indicated (*, *p* < 0.05; **, *p* < 0.01; ***, *p* < 0.001; one-way ANOVA with Tukey post hoc test).

**Figure 5 ijms-23-06972-f005:**
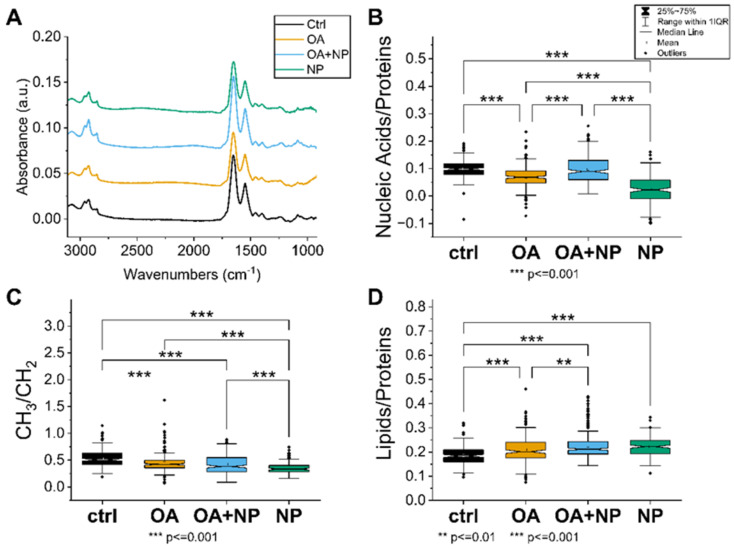
(**A**) Average spectra of the four analyzed cell groups after preprocessing, from bottom to top: Controls (CTRL) in black (average of 174 spectra), cells treated with oleic acid (OA) in yellow (average of 225 spectra), cells treated with OA+ silica-coated SPIONs (OA + nanoparticles (NP)) in blue (average of 280 spectra), and in green the average spectrum of SPION-treated cells (NP, average of 188 spectra). (**B**) Violin plot of the ratio of nucleic acid bands to total proteins. (**C**) Violin plot of the ratio of lipid bands to total amount of proteins. (**D**) Violin plot of the ratio of CH_3_ to CH_2_ bands of lipids. The same visualization parameters were used for all violin plots: CTRL in black, OA in yellow, OA + NP in blue, and NP in green. Values are sorted from the lowest value on the left to the highest on the right. Statistically significant differences in mean values are marked (**, *p* < 0.01; ***, *p* < 0.001).

**Figure 6 ijms-23-06972-f006:**
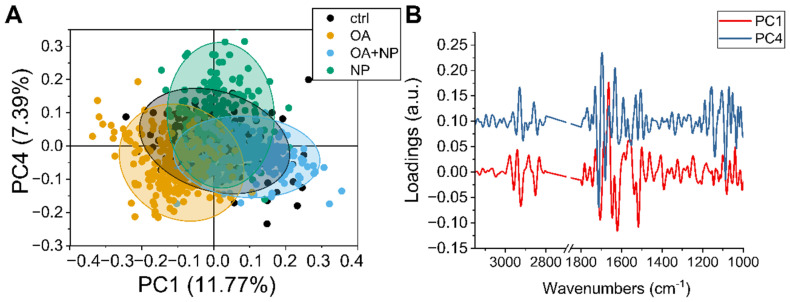
(**A**) Scatter plot of principal component analysis (PCA) scores of principal component (PC)1 and PC4. (**B**) Line plot of charge vectors corresponding to PC1 (in black) and PC4 (in red). For clarity, an offset of 0.1 a.u. was added to the y-axis.

## Data Availability

The data that support the findings of this study are available from the corresponding author upon reasonable request.

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
