# Peer review of "Oleic Acid Protects Endothelial Cells from Silica-Coated Superparamagnetic Iron Oxide Nanoparticles (SPIONs)-Induced Oxidative Stress and Cell Death"

_ijms, 2022, doi:10.3390/ijms23136972_

Round 1

Reviewer 1 Report

The manuscript “Oleic acid protects endothelial cells from superparamagnetic iron oxide nanoparticles (SPIONs)-induced oxidative stress and cell death” by Nepar et al. is well-written and describes the use of superparamagnetic iron oxide nanoparticles for induced cell death applications.

This is an original contribution which is well described and well introduced with adequate literature (even though it contains a large number of references).

I like the outline and the discussion of this study very much and would recommend minor revisions with the following stated issues:

Can you please give a reference for the following sentence:

“The silica surface enables the formation of strong covalent Si-O-Si bonds with silane molecules, which can then be further covalently bound with polyethylene glycol for colloidal stability or/and targeting agents.”

I am not sure that PEG forms covalent bonds with silanol-terminated surfaces. I also think you mean silanol and not silane (SiH4).

The same question goes for the following statement where you should also revise if it is really a silane coating and also please give a reference:

“Similar silica layers on the core of iron oxide NPs can also be made with aminosilane coatings.”

Results:

The following statement is no contradiction as maghemite also only shows one spinel phase in its XRD pattern:

“The X-ray diffraction (XRD) of the precipitated NPs showed a single spinel phase, whereas the Mossbauer spectroscopy confirmed that the SPIONs were composed of maghemite, as reported in our previous work [42].”

Figure 1: The scale bars of TEM images are too small to well evaluate the particle size. May you please increase the size of the scale bars?

Figure 2, 4, 5 and 6:

Please increase the size of text in all figures. Some are difficult and some even impossible to read.

Experimental:

For DLS and zetapotential measurements please indicate the concentration and the pH of the suspensions investigated.

Reviewer 2 Report

The aim of this study was to test the hypothesis that oleic acid has positive effect on protecting the toxicity of silica-coated SPIONs. Several experiments were performed and conducted to the final conclusion. Overall the manuscript is well in writing and the data can support their conclusion. The reviewer has the following minor comments that may help to modify the manuscript before it can be accepted for publishing.

1.      On lines 35-36, the author stated that ”Superparamagnetic iron oxide nanoparticles (SPIONs) are a group of nanoparticles (NPs) that have an iron oxide core with a size ranging from 5 nm to 15 nm” However, in this study, the average size of the NP was 19.3 ± 2.0 nm (line 116). With this size, it is doubt that the NP used in this study is superparamagnetic ION. I suggest the author to perform SQUID experiment to confirm the hysteresis loop of the used NP.  

2.      As the author stated in lines 145-152, the negative effects of SPIONs on cells are still controversial, and the results varied according to the coated materials. Since the cytotoxicity effect observed in this study cannot represent other coating situations, I suggest the author to revise the phrase SPION to “silica-coated SPIONs” throughout the manuscript.

3.      In Fig. 2E, 4C, 4G please mention the meaning of gray square bars.

4.      On Page 7, the Fig. 1 should be Fig. 3.   
